# High Intratumoral i-tRF-Gly^GCC^ Expression Predicts Short-Term Relapse and Poor Overall Survival of Colorectal Cancer Patients, Independent of the TNM Stage

**DOI:** 10.3390/biomedicines11071945

**Published:** 2023-07-08

**Authors:** Spyridon Christodoulou, Katerina Katsaraki, Panteleimon Vassiliu, Nikolaos Danias, Nikolaos Michalopoulos, Georgios Tzikos, Diamantis C. Sideris, Nikolaos Arkadopoulos

**Affiliations:** 1Fourth Department of Surgery, University General Hospital “Attikon”, National and Kapodistrian University of Athens, 12462 Athens, Greece; spyridon.christodoulou@yahoo.gr (S.C.); pant.greek@gmail.com (P.V.); ndanias@med.uoa.gr (N.D.); nmichal@med.uoa.gr (N.M.); 2Department of Biochemistry and Molecular Biology, Faculty of Biology, National and Kapodistrian University of Athens, 15701 Athens, Greece; kkatsaraki@biol.uoa.gr (K.K.); dsideris@biol.uoa.gr (D.C.S.); 3Propaedeutic Department of Surgery, University General Hospital “AHEPA”, Aristotle University of Thessaloniki, 54636 Thessaloniki, Greece; tzikos_giorgos@outlook.com

**Keywords:** colon cancer, molecular tumor markers, prognosis, prognostic biomarkers, small noncoding RNA, tRNA fragment

## Abstract

Colorectal cancer (CRC), one of the most prevalent types of cancer, requires the discovery of new tumor biomarkers for accurate patient prognosis. In this work, the prognostic value of the tRNA fragment i-tRF-Gly^GCC^ in CRC was examined. Total RNA extraction from 211 CRC patient cancer tissue specimens and 83 adjacent normal tissues was conducted. Each RNA extract was subjected to in vitro polyadenylation and reverse transcription. A real-time quantitative PCR assay was used to quantify i-tRF-Gly^GCC^ in all samples. Extensive biostatics analysis showed that i-tRF-Gly^GCC^ levels in CRC tissues were significantly lower than in matched normal colorectal tissues. Additionally, the disease-free survival (DFS) and overall survival (OS) time intervals were considerably shorter in CRC patients with high i-tRF-Gly^GCC^ expression. i-tRF-Gly^GCC^ expression maintained its prognostic value independently of other established prognostic factors, as shown by the multivariate Cox regression analysis. Additionally, survival analysis after TNM stage stratification revealed that higher i-tRF-Gly^GCC^ levels were linked to shorter DFS time intervals in patients with TNM stage II tumors, as well as an increased probability of having a worse OS for patients in TNM stage II. In conclusion, i-tRF-Gly^GCC^ has the potential to be a useful molecular tissue biomarker in CRC, independent of other clinicopathological variables.

## 1. Introduction

Colorectal cancer (CRC) is a significant issue for public health, accounting for the third most prevalent cancer diagnosis and the leading cause of cancer-related deaths globally [1]. Less than half of cases are diagnosed when the cancer is locally advanced. Nowadays, most cases are often discovered at a later stage due to the extensive use of semi-invasive endoscopic techniques and fecal blood testing, both of which have subpar diagnostic accuracy [1,2]. Despite progress in understanding the molecular and cellular underpinnings of CRC, early identification remains difficult due to the lack of symptoms in the early stages, even though the prevalence of early onset colorectal cancer, or colon cancer diagnosed in patients under the age of 50, has been rising [3,4]. Moreover, the general global incidence of CRC is anticipated to increase in the upcoming decade despite improvements in diagnostic tools and treatment approaches, highlighting the urgent need for new prognostic markers and customized therapy approaches based on molecular biomarkers [5].

The most accurate prognostic indicator for CRC patients continues to be the TNM staging system, which is based on the extent of tumor invasion depth (T), lymph node infiltration (N), and presence or absence of distant metastasis (M) [6]. Tumors that are confined to the innermost layers of the colon or rectum and have not spread to the surrounding tissues are referred to as stage I CRC [7]. Because the cancer is confined and has not yet moved past the initial site, stage I CRC has a good prognosis [8]. The migration of the tumor into adjacent tissues through the colon or rectum wall is a hallmark of stage II CRC. Due to the possibility of microscopic cancer cell spread, stage II CRC has a higher chance of recurrence than stage I [9]. Stage III CRC denotes the absence of distant metastases but the presence of local lymph node involvement. In comparison to earlier stages, stage III CRC has a higher chance of recurrence and a worse prognosis [7,9].

However, the usefulness of TNM staging in actual clinical practice, particularly in detecting high-risk stage II patients, is constrained [10]. Moreover, precise biomarkers capable of discriminating between stage II and stage III cancers and accurately predicting patient relapse are severely lacking [11,12]. Furthermore, it is critical to accurately predict patient relapse in order to inform postsurgical therapy choices and surveillance strategies. Current prognostic indicators have some predictive value but fall short of accurately predicting the chance of relapse [13]. A very promising area with several benefits is real-time prognosis using liquid samples, such as circulating tumor cells or circulating free DNA in the blood [14,15]. Nonetheless, given the especially heterogeneous nature of CRC among cancer patients, there is an urgent need for additional research to identify novel prognostic, diagnostic, and predictive biomarkers [16].

The importance of noncoding RNAs (ncRNAs) in the fine tuning of protein-coding gene expression has attracted the interest of many researchers in recent years, rendering them promising molecular biomarkers in various cancers [17,18,19,20], including CRC [21,22,23]. Besides mRNAs that compose a rich source of molecular tumor biomarkers in CRC, with the prominent examples of kallikrein-related peptidases [24,25,26,27,28] and apoptosis-related [29,30,31] or stress-induced molecules [32,33], many examples of ncRNAs, such as microRNAs (miRNAs) and circular transcripts have already been suggested as effective tumor biomarkers and/or therapeutic targets for this malignancy [34,35]. Particular miRNAs have been shown to assist diagnosis of the disease at early stages and predict patient outcomes [36,37,38,39,40], due to their implication in biological processes such as tumorigenesis, metastasis, and drug resistance [41,42].

Recently, tRNA-derived RNA fragments (tRFs) have also emerged as a novel frontier with potential diagnostic and prognostic importance among the numerous classes of ncRNAs [43,44]. These fragments are endogenous single-stranded ncRNAs ranging in length from 14 to 40 nucleotides. Notably, tRFs appear as significant regulators of gene expression acting similarly to other members of ncRNAs class, such as miRNAs. Furthermore, they post-transcriptionally regulate stability or translation of carcinogenic transcripts, leading to tumor suppression [45,46]. tRFs have been shown to significantly regulate cancer, hematologic malignancies, disorders of metabolism, inflammation, infections from viruses, and diseases of the nervous system [44]. Their expression is deregulated in cancer and hematological malignancies; this fact, along with their abundant presence in bodily fluids, renders them as molecular biomarkers in CRC [43,47]. Recent evidence suggests that particular tRFs are deregulated in this malignancy, impacting important pathways involved in cancer development. For instance, tRF-3022b regulates colorectal cell apoptosis and M2 macrophage polarization by binding to cytokines, and tRF3008A inhibits colorectal cancer development and metastasis by disrupting FOXK1 in an AGO-dependent way [48,49]. An intriguing tRF that derives from the internal part of the mature tRNA (internal tRF, i-tRF), bearing the glycine “GCC” anticodon, is i-tRF-Gly^GCC^. The oncogenic role of this tRF has already been established in ovarian cancer, where it is associated with early progression and poor overall survival, and in chronic lymphocytic leukemia, where its significance as an independent unfavorable biomarker was uncovered [50,51].

The aforementioned evidence prompted us to assess the potential of i-tRF-Gly^GCC^ expression in colorectal tumors as a prognostic molecular biomarker. To achieve this goal, we applied a real-time quantitative polymerase chain reaction (qPCR) assay for the relative quantification of i-tRF-Gly^GCC^ levels in colorectal cancer tissue specimens and adjacent normal colorectal tissues, using the comparative C_t_ method for calculations.

## 2. Materials and Methods

### 2.1. Collection of Colorectal Tissue Samples

The present study included tissue specimens from 211 patients with primary CRC, operated at the University General Hospital “Attikon”, from 2009 to 2019. All tissue samples were histologically evaluated by a pathologist and immediately frozen in liquid nitrogen. Normal colorectal tissue samples were acquired from 83 cases.

This study was approved by the Ethics Committee of the University General Hospital “Attikon” (number of approval: 13; date 29 January 2009), according to the guidelines of the Declaration of Helsinki. All patients were informed about the scope of the research and provided their consent.

### 2.2. Clinical Characteristics of CRC Patients

This study included 211 tissue samples of primary CRC and 83 adjacent normal colorectal tissue samples. In total, 108 male and 103 female CRC patients were included in this study. Patients had a median age of 66 years (interquartile range: 57–72 years) at the time of diagnosis. The clinical features of CRC patients shown in Table 1 include tumor size, histological grade, and TNM stage. According to the revised TNM classification system, patients are classified by taking into account the invasion of tumor (T), the infiltration of regional lymph nodes (N), and the potential presence of distant metastases (M) [52]. Moreover, information about treatment of CRC patients is presented in Table 1.

Survival data were available for all patients included in the current study; however, 28 of them presented distant metastasis (M1) at the time of surgery and were hence excluded from DFS analysis. The follow-up information included the date of disease recurrence diagnosis; the date and cause of death were also recorded for those patients who succumbed to their disease during the follow-up period.

### 2.3. Total RNA Extraction and Polyadenylation, Followed by First-Strand cDNA Synthesis

The DLD-1 colorectal adenocarcinoma cell line was purchased from the American Type Culture Collection (ATCC^®^) and cultured according to the ATCC guidelines. Colorectal tissue homogenization followed, and total RNA extraction was performed from DLD-1 cells and each tissue specimen using TRI Reagent^®^ (Molecular Research Center, Inc., Cincinnati, OH, USA). RNA was diluted in Storage Solution (Life Technologies Ltd., Carlsbad, CA, USA), and its concentration and purity were assessed spectrophotometrically at 260 and 280 nm with a BioSpec-nano microvolume UV–Vis spectrophotometer (Shimadju, Kyoto, Japan). All total RNA extracts were stored at −80 °C prior to their polyadenylation with *E. coli* poly(A) polymerase and reverse transcription into first-strand cDNA starting next to an oligo-dT adapter primer [53].

### 2.4. SYBR Green Based Real-Time Quantitative PCR (qPCR)

A real-time quantitative PCR method, based on SYBR Green chemistry, was applied as previously described, to perform relative quantification of i-tRF-Gly^GCC^. The comparative Ct (2^−ΔΔCt^) method was applied for all calculations to determine the tissue levels of this small ncRNA in each tissue sample [54]. *SNORD43* and *SNORD48* were used as internal reference genes to normalize the i-tRF-Gly^GCC^ expression levels; the DLD-1 cell line extract served as a calibrator in the real-time qPCR. All primers that were used were gene-specific, as previously described [51,55]. Normalized expression values of this tRF were expressed in relative quantification units (RQU).

### 2.5. Extended Biostatistics, including Disease-Free and Overall Survival Analyses

Non-parametric statistical tests were used in the biostatistics analysis. In particular, the Wilcoxon signed-rank test was used to assess the statistical significance of difference of i-tRF-Gly^GCC^ expression levels in pairs of CRC and normal adjacent tissues; differences of i-tRF-Gly^GCC^ expression levels among subgroups of patients (based on each clinicopathological factor) were checked with the Jonckheere–Terpstra test.

A receiver operating characteristic (ROC) curve was built by plotting sensitivity versus (1-specificity), and the area under the curve (AUC) was calculated. Logistic regression analysis was also performed to assess the potential of i-tRF-Gly^GCC^ expression to predict CRC occurrence.

In order to assess the prognostic value of i-tRF-Gly^GCC^ expression in CRC, we constructed Kaplan–Meier disease-free survival (DFS) and overall survival (OS) curves; for this purpose, this continuous variable was split at the median value. Stratified Kaplan–Meier survival analyses were also conducted. The differences between the curves were evaluated with the Mantel–Cox (log-rank) test. To evaluate the prognostic potential of i-tRF-Gly^GCC^ expression and determine the hazard ratio (HR) for patients’ relapse and disease-related death, bootstrapped Cox regression analyses were carried out with 1000 bootstrap samples. The bootstrap bias-corrected and accelerated (BCa) method was implemented to calculate bootstrap *p* values and 95% confidence intervals (CIs) for each estimated HR. Furthermore, multivariate prognostic models were built and adjusted for the most important clinicopathological characteristics and type of treatment each patient received. Only *p* values lower than 0.050 (*p* < 0.050) were considered as statistically significant in each statistical test.

## 3. Results

### 3.1. i-tRF-Gly^GCC^ Expression Is Downregulated in CRC Tissues, Compared to Adjacent Normal Colorectal Tissues

i-tRF-Gly^GCC^ levels in CRC specimens ranged from 0.001 to 3.4 RQU with a mean ± SEM of 0.44 ± 0.036, and from 0.002 to 1.1 RQU with a mean ± SEM of 0.36 ± 0.031 in noncancerous specimens (Table 2). Although the distribution in the two cohorts is quite similar, i-tRF-Gly^GCC^ levels were downregulated in the vast majority (58 out of 83) of the malignant tumors, compared to their matched normal tissue specimens, unraveling the utility of i-tRF-Gly^GCC^ expression levels for screening purposes (Figure 1).

### 3.2. i-tRF-Gly^GCC^ Overexpression Represents a Reliable Indicator of Poor Prognosis in CRC

A total of 61 (33.3%) out of the 183 patients who were included in the disease-free survival (DFS) analysis exhibited tumor recurrence observed during the accrual follow-up period. Similarly, 94 (44.5%) deaths associated with CRC occurred during the follow-up period. The median follow-up time was 93 months.

In order to assess the prognostic value of i-tRF-Gly^GCC^, we categorized the CRC patients into two groups, those with low i-tRF-Gly^GCC^ levels and those with higher ones, by splitting at the median the i-tRF-Gly^GCC^ expression in cancerous samples (cut-off point: 0.50 RQU). Kaplan–Meier survival analysis revealed that patients with high i-tRF-Gly^GCC^ levels have significantly shorter DFS (*p* < 0.001) and OS (*p* = 0.007) intervals, compared to patients with low i-tRF-Gly^GCC^ levels (Figure 2). These results were also confirmed by univariate Cox regression analysis, in which a hazard ratio (HR) of 2.39 (*p* < 0.001) was calculated for disease recurrence in patients with high i-tRF-Gly^GCC^ expression (Table 3), with an HR of 1.79 (*p* = 0.003) for CRC-related death in the same group of patients (Table 4), compared to CRC patients presenting with low intratumoral i-tRF-Gly^GCC^ expression.

### 3.3. The Prognostic Signficance of i-tRF-Gly^GCC^ Expression Is Independent of Other Classical Prognostic Factors Applied in CRC

In the multivariate Cox regression analysis, the importance of the i-tRF-Gly^GCC^ expression status in the prognosis of the patients’ DFS remained unaffected (HR = 2.64; *p* = 0.004), even when combined with the tumor size, histological grade, depth of tumor invasion, regional lymph node status, and treatment with radiotherapy and/or chemotherapy (Table 3). i-tRF-Gly^GCC^ retained its prognostic significance regarding OS, as well (HR = 1.56; *p* = 0.046), when combined with the aforementioned classical prognosticators plus the presence or absence of distant metastases (Table 4).

### 3.4. i-tRF-Gly^GCC^ Overexpression Predicts Tumor Recurence and Poor Prognostic Outcome in CRC Patients in TNM Stage II

After stratification, according to the most important prognostic factor used for CRC prognosis, namely, the TNM stage, patients with TNM stage II colorectal tumors overexpressing i-tRF-Gly^GCC^ showed remarkably shorter DFS intervals (*p* = 0.003) compared to patients in the same TNM stage and with low i-tRF-Gly^GCC^ levels (Figure 3A).

Furthermore, patients of TNM stage II with increased i-tRF-Gly^GCC^ levels showed an elevated probability of a poorer OS (*p* < 0.001), in comparison with patients of the same TNM stage and low i-tRF-Gly^GCC^ expression (Figure 3B). However, no statistically significant results were obtained for patients in other TNM stages.

## 4. Discussion

Given that stage I colon cancer patients have a high five-year relative survival rate, early identification of CRC is essential for patient survival. According to the American Cancer Society, this ratio lowers significantly for later-stage CRC patients. Moreover, a lot of therapeutic efforts have been directed towards developing a better and more precise classification of patients due to the variable survival outcomes and treatment responses [56]. During the last 5 years, four consensus molecular CRC subtypes (CMS) were characterized by an international collaboration of expert groups, demonstrating considerable interconnectedness between six independent classification systems [57]. Additionally, clinical researchers have focused on identifying molecular biomarkers that could potentially be employed in clinical practice for early diagnosis and reliability in CRC prognosis.

Despite being in its infancy, research on tRNA derivatives represents a research subject that has attracted scientific interest since the advent of small RNA sequencing and other innovative methods. Their involvement in numerous molecular and cellular processes that lead to the cancerous phenotype and include the dysfunctionality of transcription, cell proliferation, and differentiation has emphasized their role as potential biomarkers and therapeutic targets [58]. Furthermore, several studies have sought to link their abnormally expressed levels in cancer with the malignant phenotype [59].

It is hypothesized that tRFs, like other small ncRNAs, constitute key players in the epigenetic regulation of the protein-coding genes [60]. This observation has been proposed by several studies that showed a close association between particular tRFs and AGO or PIWI proteins in the time when gene silencing occurred in a variety of types of human specimens [61,62,63]. As a result, there is growing curiosity about their biomarker utility in solid tumors and hematological malignancies [43,51,64,65,66,67]. High-throughput genomic studies that specifically examined CRC have found significant changes in tRNA-derived small RNA levels between malignant and benign colorectal cancers [68,69]. In one of the aforementioned works, the authors discovered two distinct tRF signatures that may distinguish between colorectal adenomas and adenocarcinomas. Another study noted that 5′-tiRNA-Pro^TGG^ is associated with poor survival of CRC patients and disease recurrence [70].

The tRNA molecules bearing the glycine “GCC” anticodon represent the origin of the internal tRF investigated in this original study. The biomarker utility of the i-tRF-Gly^GCC^ molecule has been previously proposed for chronic lymphocytic leukemia and multiple myeloma [51,67], while 5′-tRF-Gly^GCC^, which derives from the same tRNAs, has been proposed as a putative discriminatory biomarker for CRC [71]. Prompted by these, in this study, we explored the biomarker potential of i-tRF-Gly^GCC^ in CRC, since it is a malignancy for which the identification of reliable biomarkers would improve patients’ prognosis and individualized treatment options [72].

Our findings showed that i-tRF-Gly^GCC^ levels were considerably lower in CRC tissues compared to paired noncancerous tissues. Considering that the majority of malignant tumors had lower levels of i-tRF-Gly^GCC^, this discovery implies that it could be used as a discriminatory biomarker for CRC. Furthermore, high i-tRF-Gly^GCC^ levels were related to a poor prognosis in CRC patients. When compared to patients with lower i-tRF-Gly^GCC^ levels, those with greater i-tRF-Gly^GCC^ levels had significantly shorter disease-free survival (DFS) and overall survival (OS) intervals. These data suggest that i-tRF-Gly^GCC^ could be used as a valid prognostic biomarker for CRC, assisting in the identification of patients at increased risk of disease recurrence and mortality from CRC.

Additionally, we investigated the prognostic utility of i-tRF-Gly^GCC^ in CRC patients according to their TNM stage. After grouping the patients based on TNM stage, we found that patients in TNM stage II had significantly shorter DFS intervals when the intratumoral i-tRF-Gly^GCC^ levels were high. Similar to this, patients with higher i-tRF-Gly^GCC^ levels and TNM stage II were more likely to have inferior OS, compared to those with TNM stage II tumors with low levels of this tRF. These results imply that i-tRF-Gly^GCC^ may be an effective prognostic marker for the course of the disease, particularly in specific stages of CRC. Patients with CRC at the TNM II stage frequently have considerably different survival probabilities. In order to develop stratification systems that more effectively define the clinical stratification and predict the course of the disease, studies have concentrated on the molecular pathways underlying CRC. CMS was developed as a first step towards the development of a new stratification system, as a recently developed stratification technique that is based on the biological characteristics of CRC patients [57]. CMS has recently been proposed by Purcell et al. since it improves the prognosis for CRC patients at the TNM II stage [73]. These data highlight the necessity for novel biomarkers, able to further stratify CRC patients, especially in groups with high heterogeneity [74].

Moreover, our findings provide significant insights both for the scientific community and regarding the clinical utility of i-tRF-Gly^GCC^. Firstly, these results add to the expanding body of evidence supporting the importance of tRNA-derived fragments as possible biomarkers in cancer. We provide valuable insights into the discriminatory and prognostic importance of the i-tRF-Gly^GCC^ fragment in CRC by particularly examining its expression levels in CRC and paired normal samples, as well as its prognostic value. This contributes to a better understanding of the complicated molecular pathways that underpin CRC formation and progression. In terms of clinical application, i-tRF-Gly^GCC^ levels in CRC patients might be measured as part of normal discriminatory procedures. The downregulation of i-tRF-Gly^GCC^ in CRC tissues implies that it could be used as a noninvasive biomarker for CRC identification and screening [75]. Furthermore, the predictive value of i-tRF-Gly^GCC^ emphasizes its usefulness in stratifying patients based on their risk of disease recurrence and CRC-related death. More specifically, incorporating i-tRF-Gly^GCC^ into existing prognostic models may improve their accuracy and allow for more precise risk categorization of CRC patients, allowing for tailored treatment methods and surveillance approaches.

Despite these promising findings, it is important to note several limitations of this study. First, although this study reveals the link between i-tRF-Gly^GCC^ levels and CRC, it does not provide mechanistic insights into the functional involvement of i-tRF-Gly^GCC^ in CRC development and progression. Understanding the underlying biological mechanisms and molecular pathways impacted by i-tRF-Gly^GCC^ would support its usage as a biomarker. In order to determine the functional significance of i-tRF-Gly^GCC^ in CRC, more functional investigations, such as in vivo tests, are required. Secondly, this study’s findings would benefit from independent confirmation in other CRC patient groups. Cross-validation across several populations and centers is critical for determining the robustness and reproducibility of the results. Furthermore, validation in cohorts with varying stages of CRC and clinical features would give a more comprehensive assessment of the biomarker’s performance. Lastly, this study focused solely on the discriminatory and prognostic potential of i-tRF-Gly^GCC^ in CRC. While the findings are encouraging, it is crucial to note that CRC is a complicated disease with various molecular alterations. A single biomarker may not reflect the whole complexities of CRC biology, limiting its clinical utility. To improve the accuracy and reliability of CRC diagnosis and prognosis, future research should consider incorporating numerous biomarkers or a panel of biomarkers.

Overall, our findings provide novel evidence for the utility of i-tRF-Gly^GCC^ as a discriminatory and prognostic biomarker in CRC. The downregulation of i-tRF-Gly^GCC^ in CRC tissues, as well as its association with a poor prognosis, emphasizes its clinical importance. Further studies should focus on verifying these findings in bigger cohorts, explaining the underlying processes, and investigating the clinical value of i-tRF-Gly^GCC^. In this challenging disease, incorporating i-tRF-Gly^GCC^ evaluation into normal CRC management has the potential to improve patient outcomes and better individualized treatment regimens.

## 5. Conclusions

According to our study, high i-tRF-Gly^GCC^ expression is associated with poor survival and higher relapse rates in CRC patients. Therefore, independent of other clinicopathological variables, the expression status of this tRF may be utilized to evaluate the prognosis of CRC patients in addition to TNM staging.

## Figures and Tables

**Figure 1 biomedicines-11-01945-f001:**
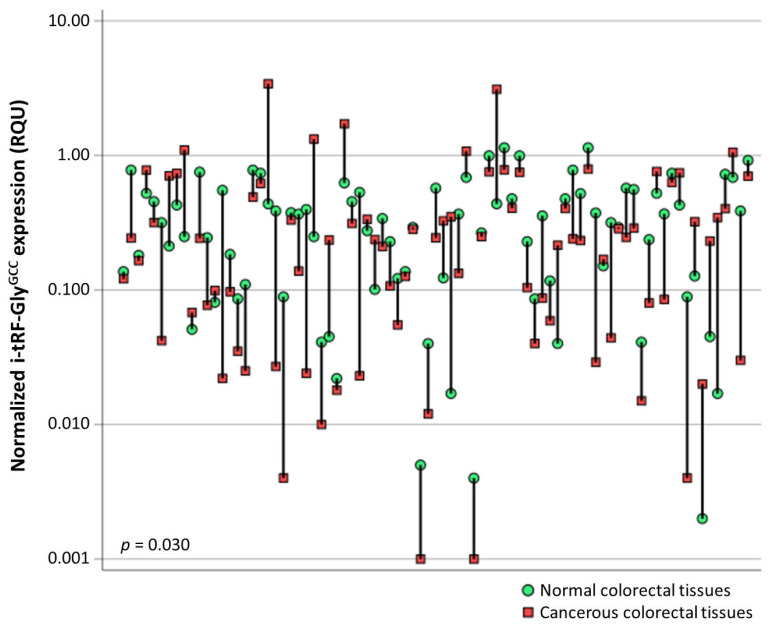
Graphical illustration of i-tRF-Gly^GCC^ expression levels in cancerous vs. normal adjacent colorectal tissues, after comparing 83 pairs of tissue specimens. The i-tRF-Gly^GCC^ expression levels were lower in most colorectal tumors. The Wilcoxon signed-rank test was used to calculate the *p* value.

**Figure 2 biomedicines-11-01945-f002:**
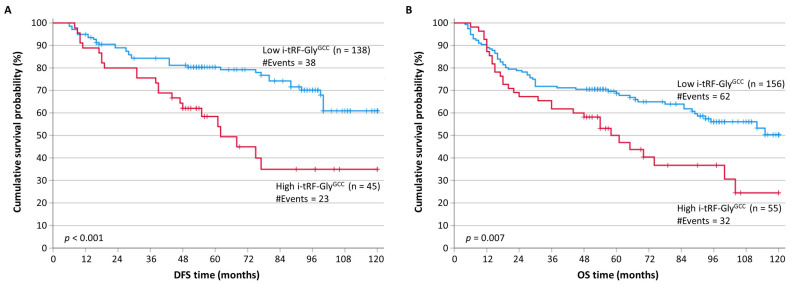
Kaplan–Meier survival curves for the disease-free survival (DFS) and overall survival (OS) of CRC patients. Patients with tumors highly expressing i-tRF-Gly^GCC^ had significantly shorter DFS (**A**) and OS (**B**) time intervals than patients bearing tumors with low i-tRF-Gly^GCC^ levels. The *p* values were calculated using the Mantel–Cox (log-rank) test.

**Figure 3 biomedicines-11-01945-f003:**
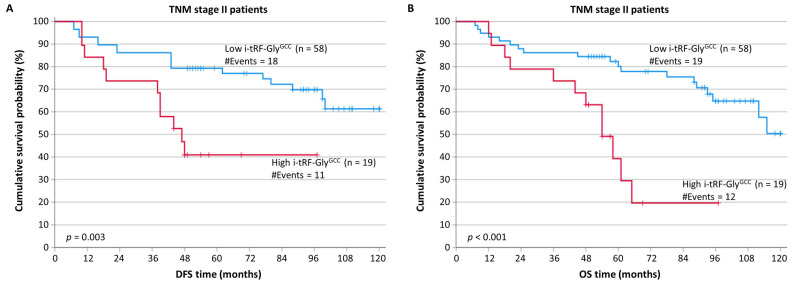
Stratified Kaplan–Meier survival curves for the disease-free survival (DFS) and overall survival (OS) of CRC patients, according to TNM stage. TNM stage II patients with tumors highly expressing i-tRF-Gly^GCC^ had shorter DFS (**A**) and OS (**B**) time intervals than patients bearing tumors with low i-tRF-Gly^GCC^ levels. The *p* values were calculated using the Mantel–Cox (log-rank) test.

**Table 1 biomedicines-11-01945-t001:** Clinical characteristics and treatment of CRC patients included in the current study.

	Number of Patients (%)
Histological grade	
I	26 (12.3%)
II	154 (73.0%)
III	31 (14.7%)
T (tumor invasion)	
T1	9 (4.3%)
T2	37 (17.5%)
T3	116 (55.0%)
T4	49 (23.2%)
N (nodal status)	
N0	123 (58.3%)
N1	60 (28.4%)
N2	28 (13.3%)
M (distant metastasis)	
M0	183 (86.7%)
M1	28 (13.3%)
TNM stage	
I	40 (19.0%)
II	77 (36.5%)
III	66 (31.3%)
IV	28 (13.2%)
Treatment with radiotherapy (207/211 patients)	
No	165 (79.7%)
Yes	42 (20.3%)
Treatment with chemotherapy (207/211 patients)	
No	83 (40.1%)
Yes	124 (59.9%)

Abbreviation: TNM: tumor, node, and metastasis.

**Table 2 biomedicines-11-01945-t002:** Descriptive statistics of i-tRF-Gly^GCC^ expression levels and other features of CRC patients.

Variable	Mean ± SEM	Range	Quartiles		
			1st	2nd (Median)	3rd
Normalized i-tRF-Gly^GCC^ expression (RQU)					
in cancerous tissues (*n* = 211)	0.44 ± 0.036	0.001–3.4	0.11	0.30	0.59
in normal tissues (*n* = 83)	0.36 ± 0.031	0.002–1.1	0.12	0.32	0.52
Patient age (years)	65 ± 0.8	35–93	57	66	72
Tumor size (cm^2^)	19.3 ± 1.1	0.8–132	9.8	14.0	24.0

Abbreviations: RQU, relative quantification units; SEM, standard error of the mean.

**Table 3 biomedicines-11-01945-t003:** i-tRF-Gly^GCC^ expression and disease-free survival (DFS) of CRC patients.

	Univariate Analysis (*n* = 183)			Multivariate Analysis (*n* = 181)		
Covariate	HR ^1^	BCa ^2^ 95% CI ^3^	*p* Value ^4^	HR ^1^	BCa ^2^ 95% CI ^3^	*p* Value ^4^
i-tRF-Gly^GCC^ expression (high vs. low)	2.39	1.40–4.29	*<0.001*	2.64	1.45–5.33	*0.004*
Tumor size	0.99	0.96–1.00	0.13			
Histological grade (ordinal)	2.51	1.46–4.54	*0.003*	2.20	1.13–4.80	*0.027*
T (ordinal)	1.69	1.22–2.59	*0.008*	1.59	0.97–2.81	0.065
N (ordinal)	1.47	0.98–2.16	0.051	1.06	0.61–1.78	0.82
Treatment with radiotherapy (yes vs. no)	1.34	0.72–2.39	0.36	0.84	0.38–1.61	0.60
Treatment with chemotherapy (yes vs. no)	1.72	0.99–3.33	0.062	1.17	0.55–2.50	0.69

^1^ Hazard ratio, estimated from proportional hazard Cox regression. ^2^ Bias-corrected and accelerated. ^3^ Confidence interval of the estimated HR. ^4^ Statistically significant bootstrap *p* values are shown in italics.

**Table 4 biomedicines-11-01945-t004:** i-tRF-Gly^GCC^ expression and overall survival (OS) of CRC patients.

	Univariate Analysis (*n* = 211)			Multivariate Analysis (*n* = 207)		
Covariate	HR ^1^	BCa ^2^ 95% CI ^3^	*p* Value ^4^	HR ^1^	BCa ^2^ 95% CI ^3^	*p* Value ^4^
i-tRF-Gly^GCC^ expression (high vs. low)	1.79	1.18–2.68	*0.003*	1.56	0.98–2.43	*0.046*
Tumor size	1.00	0.99–1.02	0.53			
Histological grade (ordinal)	1.91	1.21–3.04	*0.008*	1.24	0.73–2.07	0.41
T (ordinal)	1.88	1.34–2.74	*0.002*	1.42	0.93–2.31	0.12
N (ordinal)	1.99	1.48–2.75	*<0.001*	1.46	0.96–2.14	*0.047*
M (M1 vs. M0)	7.17	4.13–13.52	*<0.001*	4.71	2.21–12.48	*<0.001*
Treatment with radiotherapy (yes vs. no)	0.94	0.53–1.54	0.84	1.16	0.57–2.08	0.63
Treatment with chemotherapy (yes vs. no)	1.15	0.75–1.86	0.53	0.57	0.33–0.98	*0.033*

^1^ Hazard ratio, estimated from proportional hazard Cox regression. ^2^ Bias-corrected and accelerated. ^3^ Confidence interval of the estimated HR. ^4^ Statistically significant bootstrap *p* values are shown in italics.

## Data Availability

The data presented in this study are available on reasonable request from the corresponding authors.

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
