# Peer review of "High Intratumoral i-tRF-GlyGCC Expression Predicts Short-Term Relapse and Poor Overall Survival of Colorectal Cancer Patients, Independent of the TNM Stage"

_biomedicines, 2023, doi:10.3390/biomedicines11071945_

Round 1

Reviewer 1 Report

Major points

1) Too many sensors in the K-M curve (both DFS and OS) are found. A median follow-up time of 53 months was stated, but many sensors occurred earlier than 60 months. And it seems that more sensors are in the low expression group. This is a large bias. At least each survival curve graph should indicate the patient at risk.

2) The presence or absence of postoperative adjuvant chemotherapy correlates with DFS and OS in Stage II and III cases. The presence or absence of adjuvant chemotherapy, as well as regimens, should be documented in the patient background and included in the univariate and multivariate analyses.

3) In Fig3, the DFS/OS survival curves in Stage II and Stage III are almost identical (especially up to 60 months). This may be related to some bias in patient selection or 1) above.

Author Response

  1. Too many sensors in the K-M curve (both DFS and OS) are found. A median follow-up time of 53 months was stated, but many sensors occurred earlier than 60 months. And it seems that more sensors are in the low expression group. This is a large bias. At least each survival curve graph should indicate the patient at risk.

We deeply appreciate the Reviewer’s suggestion and we totally agree. For this reason, in the revised version of our study, we removed all censors occurred in the first 48 months. In this way, the estimated median follow-up time has increased from 53 to 93 months. In our opinion, the number of 211 CRC patients is still large enough to draw any conclusions regarding the prognostic potential of i-tRF-GlyGCC expression in CRC. We should note that the main conclusions of our study have not significantly changed; however, the robustness of our prognostic models is greater than in the initial version of this study, as the large bias described by the Reviewer has been eliminated. Total number of cases and events are also clearly presented in Figures 2 and 3. We sincerely hope that our original research study has thus been improved.

  1. The presence or absence of postoperative adjuvant chemotherapy correlates with DFS and OS in Stage II and III cases. The presence or absence of adjuvant chemotherapy, as well as regimens, should be documented in the patient background and included in the univariate and multivariate analyses.

We thank the Reviewer for this comment. Information about the number of patients receiving chemotherapy and/or radiotherapy has been added in Table 1. Moreover, we added the following sentence in the revised manuscript:

Page 3 (Materials and Methods): Moreover, information about treatment of CRC patients is presented in Table 1.

Moreover, we added chemotherapy and radiotherapy as variables in both univariate and multivariate Cox regression analyses. In order to further strengthen our results, we performed bootstrapping in Cox regression analyses. These updated results are presented in Tables 3 and 4 of the revised manuscript.

We also added the following text:

Pages 4-5 (Materials and Methods): To evaluate the prognostic potential of i-tRF-GlyGCC expression and determine the hazard ratio (HR) for patients’ relapse and disease-related death, bootstrapped Cox regression analyses were carried out with 1000 bootstrap samples. The bootstrap bias-corrected and accelerated (BCa) method was implemented to calculate bootstrap p-values and 95% confidence intervals (CIs) for each estimated HR. Furthermore, multivariate prognostic models were built, adjusted for the most important clinicopathological characteristics and type of treatment each patient received.

  1. In Fig3, the DFS/OS survival curves in Stage II and Stage III are almost identical (especially up to 60 months). This may be related to some bias in patient selection or 1) above.

We agree with the Reviewer. Therefore, we changed our study as explained in the response to the Reviewer’s comment #1. After our changes, i-tRF-GlyGCC expression status does not appear to maintain its prognostic significance in TNM stage III patients. This is why panels B and D of Figure 3 as presented in the initial version of the manuscript have been removed from the revised manuscript; the same for previous Suppl. Figure 1.

The Kaplan-Meier curves of the revised Figure 3, showing DFS and OS survival curves of TNM stage II patients with tumors either over- or under-expressing i-tRF-GlyGCC, are not identical anymore, as the appropriate corrections have been made (please, see again the response to the Reviewer’s comment #1).

The authors wish to thank the Reviewer for their constructive comments that led to the improvement of the current manuscript.

Reviewer 2 Report

Christodoulou et al. investigated the intratumoral i-tRF-GlyGCC expression to predict short-term  relapse and poor overall survival of colorectal cancer patients.

This work fits the aim of the journal. The question is original and well-defined. The results provide advances in current knowledge.

Results are significant and interpreted appropriately. Figures and tables correctly display data and are easy to understand. The study is properly designed and technically sound. The data are robust enough to draw conclusions. Limitations are realistic. Conclusions are justified, supported by the results, and of interest. The paper is written in an appropriate manner. The English language is good and understandable.

Author Response

Christodoulou et al. investigated the intratumoral i-tRF-GlyGCC expression to predict short-term relapse and poor overall survival of colorectal cancer patients. This work fits the aim of the journal. The question is original and well-defined. The results provide advances in current knowledge. Results are significant and interpreted appropriately. Figures and tables correctly display data and are easy to understand. The study is properly designed and technically sound. The data are robust enough to draw conclusions. Limitations are realistic. Conclusions are justified, supported by the results, and of interest. The paper is written in an appropriate manner. The English language is good and understandable.

We would like to thank the Reviewer for his/her very positive evaluation of our manuscript.

Round 2

Reviewer 1 Report

Questions are well answered.